# One-Year Impact of Occupational Exposure to Polycyclic Aromatic Hydrocarbons on Sperm Quality

**DOI:** 10.3390/antiox13101181

**Published:** 2024-09-29

**Authors:** Mª Victoria Peña-García, Mª José Moyano-Gallego, Sara Gómez-Melero, Rafael Molero-Payán, Fernando Rodríguez-Cantalejo, Javier Caballero-Villarraso

**Affiliations:** 1Clinical Analyses Service, Reina Sofía University Hospital, 14004 Córdoba, Spain; victoria.penagarcia@gmail.com (M.V.P.-G.); mjmoyanog77@yahoo.es (M.J.M.-G.); fernando.rodriguez.c.sspa@juntadeandalucia.es (F.R.-C.); 2Maimónides Biomedical Research Institute of Córdoba (IMIBIC), 14004 Córdoba, Spain; sara.gomez.melero@gmail.com (S.G.-M.); molerorafael@gmail.com (R.M.-P.); 3Department of Biochemistry and Molecular Biology, Universidad of Córdoba, 14071 Córdoba, Spain

**Keywords:** polycyclic aromatic hydrocarbons (PAHs), occupational health, solar thermal plant, fertility, sterility, sperm quality, oxidative stress

## Abstract

Background: Polycyclic aromatic hydrocarbons (PAHs) have toxic potential, especially as carcinogens, neurotoxins, and endocrine disruptors. The objective of this study is to know the impact of exposure to PAHs on the reproductive health of male workers who operate in solar thermal plants. Methods: Case–control study. A total of 61 men were included: 32 workers exposed to PAH at a solar thermal plant and 29 unexposed people. Seminal quality was studied both at the cellular level (quantity and quality of sperm) and at the biochemical level (magnitudes of oxidative stress in seminal plasma). Results: In exposure to PAHs, a significantly higher seminal leukocyte infiltration was observed, as well as lower activity in seminal plasma of superoxide dismutase (SOD) and a reduced glutathione/oxidised glutathione (GSH/GSSG) ratio. The oxidative stress parameters of seminal plasma did not show a relationship with sperm cellularity, neither in those exposed nor in those not exposed to PAH. Conclusion: One year of exposure to PAH in a solar thermal plant does not have a negative impact on the sperm cellularity of the worker, either quantitatively (sperm count) or qualitatively (motility, vitality, morphology, or cellular DNA fragmentation). However, PAH exposure is associated with lower antioxidant capacity and higher leukocyte infiltration in seminal plasma.

## 1. Introduction

Infertility is a common health problem brought on by a disorder of the male or female reproductive system. It is defined as the inability to achieve pregnancy after 12 months or more of regular sexual intercourse in the absence of contraceptive measures. Worldwide, it affects 1 in 6 couples (of which male factors account for almost half of the cases), involving 48 million couples and 186 million people. The prevalence of infertility has increased in recent years due to several factors: (i) chronological delay of first pregnancy; (ii) alterations in semen quality due to habits such as smoking and/or alcohol consumption, or exposure to environmental toxic substances such as polycyclic aromatic hydrocarbons (PAHs), which alter male reproductive function affecting seminal quality [1,2,3].

These PAHs are a large group of chemical compounds and are polymeric derivatives of benzene. PAHs are ubiquitous and therefore represent a source of constant human exposure to these compounds, which have toxic potential at different levels in the body. Such exposure can come from natural sources (such as forest fires or volcanic eruptions), or human-made sources (such as domestic heating, vehicle traffic, or even smoking, as well as excessive consumption of smoked foods) [4,5,6].

These chemicals are of great importance for human health because of their toxic potential, especially as carcinogens, neurotoxicants, and endocrine disruptors. Furthermore, they are promutagenic elements. One of the mechanisms of action of PAHs is by inducing oxidative and nitrosative stress. Recent studies have shown that PAHs increase oxidative stress in seminal plasma and are capable of causing extensive damage to sperm DNA. This can lead to significant changes in the number, concentration, motility, and morphology of spermatozoa, which are highly susceptible to oxidation due to the amount of membrane-unsaturated fatty acids and the lack of cytoplasmic antioxidant enzymes [6,7,8]. The overall consequence would be a decrease in sperm quality, with subsequent compromise to male fertility [9].

PAHs are, basically, formed when organic matter in general, such as coal, wood, tobacco, or vegetation, is subjected to high temperatures for a sufficient time [10]. A proposed scheme for the carcinogenic potential of environmental exposure considers the following stages: exposure, metabolic activation, formation of compounds between PAHs and cellular DNA, mutations in critical genes such as p53 (tumour repressor gene), and a succession of cascade mutations in other genes. The biotransformation of PAHs involves a series of enzymes that catalyse oxidation, reduction and hydrolysis reactions (cytochrome P-450-CYP enzymes), and enzymes that catalyse conjugation reactions (sulfotransferase, epoxide hydrolase, glutathione-S-transferase, and UDP-glycosyltransferase). These enzyme systems are distributed in all tissues of the body. The enzymes responsible for the metabolic activation of PAHs, including benzopyrene, are CYP1A1, CYP1B1, and, to a lesser extent, CYP1A2 in conjunction with epoxide hydrolase. These two enzymes are widely distributed in the lungs, although they are also observed to a lesser extent in other tissues. Because PAHs are ubiquitous, it would be desirable for legislation to impose maximum levels for the presence of these carcinogens in various environments. The main routes of entry of these substances into the body are inhalation and dermal routes. Although less frequent, another possible route of contact is the digestive tract, because of ingestion of food containing these molecules even at very low concentrations (less than 1%) [10,11,12].

Today, the main sources of PAHs are exhaust from cars, aircraft, ships, electric power generation plants, waste incinerators, sheating of buildings, forest fires, and tobacco smoke, as well as smoked, grilled, or barbecued food [2,3]. Occupational sources of PAHs, besides coal, tar, and asphalt, are soot, creosote, and mineral oils (lubricating oils). A special mention should be made of electricity generation plants, in particular solar thermal power plants. The presence of PAHs in solar thermal platforms is because of the thermal fluid used to transfer heat from the solar plate to the power block, which suffers progressive degradation with time and use. The leakage and degradation of these oils form certain toxic species, including different types of PAHs [2,3,7,8]. Subsequently, other consequences of PAHs have been described, such as neurotoxicity and the impact on male and female reproductive health (subfertility/infertility) [4,6,13]. Regarding the latter consequence, various types of PAHs are considered toxic agents at the gonadotropic level, whose pathogenic mechanisms can manifest themselves on three levels: (i) gonadal dysfunction (by direct toxicity to the gonad); (ii) gamete disruption; and (iii) endocrine disruption (with potential multifaceted effects: libido disorders, sexual behaviour disorders, and alterations of the hypothalamic–pituitary–gonadal axis, even compromising gametogenesis) [4,9,10].

Continuous contact with PAHs results in increased oxidative stress in seminal plasma, due to an excess of free radicals and a lack of antioxidants to counteract them. This damages the spermatozoa once they are ejaculated, due to the generation of large amounts of reactive oxygen species (ROS). These ROS cause damage to the sperm membrane, decrease motility, and cause alterations to vital structures such as cytoplasmic components, mainly to the genetic material (DNA) [14,15].

However, small amounts of ROS are necessary to maintain normal sperm function without overwhelming the sperm’s antioxidant system [16]. Therefore, the determination of oxidative stress or the presence of ROS in ejaculated spermatozoa can be a diagnostic tool to consider in the identification of the aetiology of male infertility [14,15,17,18].

Antioxidants are present in both seminal plasma and sperm that protect gonadal cells and mature sperm from oxidative damage. Free radicals can directly damage sperm DNA. Under physiological conditions, sperm DNA is packaged by protamines that protect it from free radical attack. These free radicals can initiate programmed cell death (apoptosis) within the spermatozoon, leading to enzymatic degradation of DNA [19,20]. In short, we may classify antioxidation mechanisms into three types: (i) primary defences (prevent free radicals from forming); (ii) secondary defences (inactivate free radicals already formed); and (iii) tertiary defences (repair oxidative damage to DNA) [4,5,6,13,21].

Some molecules are involved in oxidative stress, both pro-oxidant and antioxidant mechanisms. Within the latter, we distinguish between enzymatic and non-enzymatic antioxidants. Among the enzymatic antioxidants, we can highlight superoxide dismutase (SOD), catalase, glutathione peroxidase (GPX), and glutathione-s-transferase (GST). Non-enzymatic antioxidants include glutathione, vitamin A, vitamin C, vitamin E, and coenzyme Q10 [7,14,22]. The role of catalase (belonging to the oxidoreductase family) is worth mentioning. Also, special mention should be made of SOD, which produces the dismutation of superoxide anion into oxygen and hydrogen peroxide. Because of this, it is an important antioxidant defence in most cells [16,19,23].

Therefore, in work environments with repeated and/or abundant exposure to PAHs, primary prevention of worker exposure by means of PPE (personal protective equipment) and regular monitoring of work facilities is essential. In addition, early diagnosis of diseases related to PAHs is necessary. However, knowing that the nature of the metabolites resulting from the degradation (due to temperature and/or age) of thermal fluids is not chemically well characterised, it is very difficult to consider monitoring environmental exposure at a collective level by determining toxinss in the installations [4,6,7,8]. For these reasons, close monitoring of a broad spectrum of symptoms and signs (clinical, analytical, and imaging) related to the toxicities mentioned above would be desirable in order to allow for the earliest possible diagnosis. The present study aims to determine the impact of PAHs on male reproductive health in workers operating in solar thermal power plants.

## 2. Materials and Methods

A cross-sectional, comparative study between subjects exposed and not exposed to PAHs was carried out. The exposed group was workers at a solar thermal platform who had been in their jobs for just one year. At their workstations, there were two areas of exposure, high and low. The areas of high exposure were the areas with the solar panels and the areas of low exposure were the hangars where the control rooms were located. The workers had a shift schedule, had a rotating workstation, and changed their workplace every three months. This way, the workers had the same number of hours of exposure over time. The high-exposure areas were considered as such by the company itself, depending on the amount of combustion vapours from the heating of the heat transfer fluid (HTF), which was a molten-salt-based compound. The non-exposed group included volunteers with similar characteristics in terms of age and anthropometric profile, all of whom were male. They had other jobs not related to exposure to fuels or pollutants and lived in towns in the region far from the solar thermal plant. We asked all of them (both groups) about tobacco use and found 3 smokers in the exposed group and 5 in the unexposed group; we considered the number to be insignificant to draw inferences. None of the participants were accepted if they had previously been vasectomised or had previous reproductive health problems.

Sperm samples were collected in the laboratory by masturbation after 3–4 days of sexual abstinence. Sperm analyses were carried out after complete liquefaction at 37 °C. A spermiogram was performed in the andrology laboratory of Reina Sofía University Hospital (Córdoba, Spain) and evaluated in accordance with the 2010 World Health Organization guidelines [24]. After centrifugation (2500× *g*) for 10 min in a centrifuge cooled to 4 °C, seminal plasma was separated carefully. During the assay period, samples were stored at −80 °C until analysis in trace element-free tubes (less than 30 days). Seminal plasma samples were analysed in duplicate.

Sperm quality was studied at three levels: (i) Biochemical analysis of pro-oxidant–antioxidant balance parameters in seminal plasma: superoxide dismutase (SOD), lipoperoxides (LPO), reduced glutathione (GSH), oxidised glutathione (GSSG), GSH/GSSG ratio, and nitrite (NO); all of them were analysed using Bioxytech S.A. kits (Oxis International^®^, Portland, OR, USA) and by following the manufacturer’s recommendations. (ii) Spermiogram: ejaculate volume, viscosity, sperm concentration, motility, sperm progression, morphology, and vitality. The latter two parameters were examined by means of eosin–nigrosin staining after spreading 20 µL of sperm sample onto a slide. An Olympus phase-contrast microscope CX41 (Olympus^®^, Tokyo, Japan) was used. (iii) Sperm DNA fragmentation test (Halosperm^®^, HalotechDNA^®^, Madrid, Spain): indicates what percentage of spermatozoa have damaged genetic material and complements the information provided by the spermiogram. The research was based on sperm chromatin dispersion (SCD) and used 25 20 µL semen samples (Figure 1). To measure the dynamic parameters of spermatozoa (motility and progression) and DNA fragmentation, a CEROS II computer-aided sperm analysis system (Hamilton Thorne^®^, Beverly, MA, USA) was used. This system was coupled to an Olympus phase-contrast BHS-microscope BH2 (Olympus^®^, Tokyo, Japan). It used 20 µL aliquots of semen onto the ‘Makler Counting Chamber’.

Various clinical variables were also considered: age, weight, height, body mass index (BMI), tobacco or alcohol consumption, presence of any health problems (hypertension, dyslipidaemia, diabetes, depression), and possible treatments.

### 2.1. Statistical Analysis

A Shapiro–Wilk test was performed to observe normality for each of the study variables. For variables that followed a normal distribution, in the inter-group study, the parametric Student’s *t*-test was used for the comparison of continuous quantitative variables between PAH-exposed and non-exposed groups, or the non-parametric Mann–Whitney U-test in case of non-normality. The chi-square test was used to compare the frequencies of qualitative variables. Prior to the comparison of inter-group variables, the homogeneity of variances was checked using Snedecor’s F-test. In the intra-group study, Pearson correlations were performed to determine the potential influence of each seminal plasma oxidative stress variable on each fertilisation capacity variable (i.e., the different sperm characteristics). Data analysis was conducted with the SPSS application (SPSS INC. Version 25 for Windows). A statistically significant difference was considered to exist if a value of *p* < 0.05 was obtained.

### 2.2. Ethical Concerns

The standards of good clinical practice and the principles set out in the Declaration of Helsinki were considered. Data were anonymised in accordance with Regulation (EU) 2016/679 of the European Parliament and Organic Law 3/2018 on the protection of personal data and guarantee of digital rights. Law 14/2007 on biomedical research and Law 31/1995 on the prevention of occupational hazards and risks related to exposure to biological agents were respected. All participants signed an informed consent form. The Research Ethics Committee of our hospital approved the conduct of this study.

## 3. Results

A total of 61 male subjects, aged 25–53 years, were included. Of these, 32 belonged to the group of workers exposed to PAHs in the solar thermal plant and 29 were people not exposed to PAHs. A comparison of the values of the variables considered was made between the two groups. The results of this inter-group study are shown in Table 1 and Table 2.

Significant differences were observed in total sperm count and seminal leukocyte infiltration, as well as in SOD levels and GSH/GSSG ratios (Figure 2). The latter were lower in the exposed group, while leukocyte infiltration was higher in the exposed group.

Intra-group studies were also performed in both PAH-exposed and non-exposed individuals. Within each group, associations were sought between the 15 parameters assessed in the spermiogram and the 4 variables of seminal oxidative stress analysed.

Pearson correlations were performed between age and the different parameters of seminal oxidative stress, between age and the percentage of sperm DNA fragmentation, and between the total sperm count, the spermatozoa percentages according to qualitative assessment (motility, vitality, and morphology), and the different magnitudes of oxidative stress in seminal plasma. However, no relationship was observed between any of the variables mentioned, neither in the exposed group nor in the non-exposed group.

## 4. Discussion

In relation to the results of the present study, it can be affirmed that one year of occupational exposure to PAHs by workers at a solar thermal plant does not imply a limitation in male fertility in terms of the parameters considered by a standard spermiogram. Firstly, it should be noted that the total sperm count (count in mL per ejaculated sperm volume) was not reduced in the group of exposed workers.

If we consider the qualitative variables (vitality, motility, morphology), as well as the SFA variable (DNA fragmentation measured by means of Halosperm), we did not find statistically significant differences between the exposed and unexposed groups. Although our study also found no positive correlation between age and decreased sperm fertilisation capacity, some authors argue that ageing is associated with the accumulation of reactive oxygen species in seminal plasma and other body fluids [25,26,27]. We can justify the findings of our study by considering that the subjects studied were not really old males, as the mean age was 36 years. Thus, the consequences associated with ‘ageing’ would not yet be denoted at the ages of the subjects examined here.

However, two antioxidant parameters, the activity of SOD and the GSH/GSSG ratio, were found to be decreased in the exposed group. This could reflect that there is a decrease in antioxidant substances in seminal plasma, which might translate into increased oxidative stress in the sperm of PAH-exposed individuals [27,28,29,30]. Numerous studies have also confirmed that oxidative stress is a factor in telomere shortening and dysfunction. Interestingly, while the cellular and molecular characteristics of spermatozoa make them more susceptible to oxidative damage than any other cell type, they are also the only cell type in which telomere lengthening accompanies ageing [29]. It should be noted that in our study no differences were observed in sperm DNA (in terms of sperm DNA fragmentation), whether in relation to seminal oxidative stress parameters in either group or when comparing the percentage of fragmented DNA between the two groups considered. However, a higher sperm leukocyte infiltration and abnormal sperm morphology were observed in the exposed group, which may be related to the higher pro-oxidant status mentioned above [31,32,33,34].

None of the oxidative stress molecules studied here have been shown to have an impact on the various parameters considered in a standard spermiogram, either in quantitative or qualitative terms. Many studies confirm that abnormal spermatozoa often show the typical features of oxidative stress, i.e., an excessive level of reactive oxygen species (ROS) and a reduced fertilisation capacity [31,35,36]. The cause of sperm abnormality and the presence of ROS may be because these molecules can promote oxidative changes in membranes, including a loss of membrane integrity and fluidity [37]. The plasma membrane of spermatozoa is highly susceptible to lipid peroxidation given its high levels of polyunsaturated fatty acids. These circumstances, together with the capacity of these cells to generate free radicals, make them particularly sensitive to oxidative damage [38,39].

In the present study, the biochemical parameters related to oxidative stress that were studied in seminal plasma were not analysed in peripheral blood. Some authors suggest the existence of a relationship between body oxidative status (analysing blood or urine) and sperm status [7,40]. However, other studies do not observe such a relationship [28,35,36].

It should be considered that the changes observed correspond to one year of occupational exposure. Consequently, we cannot affirm that PAH exposure does not interfere in any way with male fertility over a longer period of time [28,41,42].

If, despite the above, we consider the proposal that there is a link between oxidative stress and male infertility, we should insist on the implementation of primary prevention measures through personal protective equipment (PPE), as well as secondary prevention measures. Among these, we could cite an accurate detection method to measure ROS content that is easily implementable in an andrology laboratory, such as the use of fluorescent probes attachable to sperm cells, which are detected by flow cytometry or fluorescent microscopy [34].

Regarding the weaknesses and strengths of this study, we can point out some study limitations. Lifestyle aspects that may be related to sperm quality, such as dietary habits, were not measured. A possible inconsistency may be that for the inter-subject variability that exists in the general population in some spermiogram parameters (such as total cell count), the overall samples studied are not large in size. Given that exposure time was only one year, and the study involved a small group, this study should be considered as preliminary. This group size may be the reason, for example, for the high standard deviation of the fragmentation and degradation of the spermatozoa, so these results should be interpreted with caution. However, it should be noted that the exposed group included the entire workforce of the solar thermal plant; all workers had the same length of exposure to PAHs because the company was inaugurated a year earlier, and all were new to their jobs.

Future research should focus on the measurement of PAHs and other metabolites related to the degradation of heat transfer fluids (HTFs) in order to measure them in ambient air. Correlations between these environmental levels and possible changes in the health of exposed individuals could then be studied. It could also be useful to identify the presence of these pollutants in biological samples. For future research, we propose studies with larger sample sizes, even if this implies multicentre approaches, as well as longer follow-up periods.

## 5. Conclusions

After one year of exposure to PAHs in a solar thermal plant, no negative impact was observed on the workers’ sperm cellularity, neither at a quantitative level (sperm count), nor at a qualitative level (motility, vitality, morphology, or cellular DNA fragmentation). However, PAH exposure is associated with lower antioxidant capacity and higher leukocyte infiltration in seminal plasma.

## Figures and Tables

**Figure 1 antioxidants-13-01181-f001:**
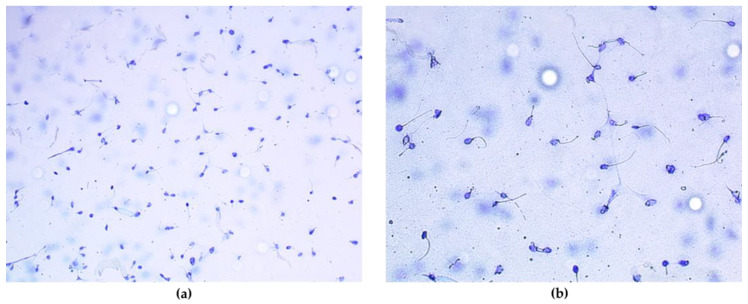
Images of sperm DNA fragmentation. The halo indicates DNA integrity. Magnified 20× (**a**). Magnified 40× (**b**).

**Figure 2 antioxidants-13-01181-f002:**
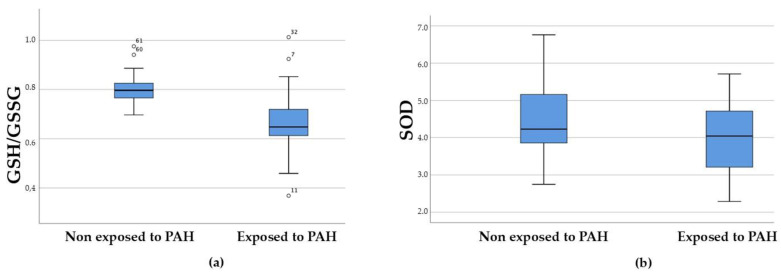
Seminal plasma biomarker levels shown comparatively in the non-exposed and exposed groups. GHS/GSSG ratio (**a**). SOD (**b**).

**Table 1 antioxidants-13-01181-t001:** Inter-group study: spermiogram parameters.

Parameter	Exposed to PAHs (N = 32) $\bar{x}$ ± SD	Not exposed to PAHs (N = 29) $\bar{x}$ ± SD	*p* Value
Age (years)	37.23 ± 8.19	35.77 ± 8.50	0.499
Volume (mL)	3.96 ± 1.37	3.40 ± 1.34	0.113
pH	8.17 ± 0.29	8.16 ± 0.233	0.817
Density (10^6^/mL)	165.74 ± 11.64	121.01 ± 131.08	0.167
Total sperm count (ml)	588.00 ± 376.44	598.53 ± 261.37	0.061
% Type A	28.22 ± 16.35	30.14 ± 20.52	0.686
% Type B	24.03 ± 15.78	12.66 ± 7.94	0.060
A and B	52.25 ± 20.71	42.79 ± 23.45	0.100
% immotile	45.38 ± 20.76	52.52 ± 27.88	0.258
% living	74.91 ± 11.93	74.14 ± 13.57	0.815
% dead	25.09 ± 11.93	25.86 ± 13.57	0.816
Normal morphology (%)	5.59 ± 4.18	7.41 ± 3.64	0.076
Abnormal morphology (%)	94.59 ± 4.20	92.59 ± 3.64	0.052
Fragmented + Degraded (mL)	330.66 ± 498.93	234, 86 ± 110.72	0.316
SDF%	0.52 ± 0.14	0.57 ± 0.18	0.253
leukocytes/20 fields	4.47 ± 4.79	1.45 ± 1.66	0.002 *

* *p* < 0.05; SDF: sperm DNA fragmentation.

**Table 2 antioxidants-13-01181-t002:** Inter-group study: oxidative stress in seminal plasma.

Parameter	Exposed to PAHs (N = 32) $\bar{x}$ ± SD	Not exposed to PAHs (N = 29) $\bar{x}$ ± SD	*p* Value
SOD (U/mL)	4.00 ± 0.85	4.47 ± 0.94	0.048 *
LPO (nmol/mg protein)	3.77 ± 0.66	3.95 ± 0.87	0.377
GSH/GSSG (nmol/mg protein)	0.66 ± 0.12	0.80 ± 0.06	0.000 *
NO (nmol/mg protein)	4.22 ± 0.84	4.13 ± 0.86	0.658

* *p* < 0.05; SOD: superoxide dismutase; LPO: lipid peroxide; GSH/GSSG: reduced/oxidised glutathione ratio; NO: nitrite.

## Data Availability

Data are contained within the article.

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
