# Peer review of "One-Year Impact of Occupational Exposure to Polycyclic Aromatic Hydrocarbons on Sperm Quality"

_antioxidants, 2024, doi:10.3390/antiox13101181_

Round 1

Reviewer 1 Report

The paper entitled “Impact of Occupational Exposure to Polycyclic Aromatic Hydrocarbons on Sperm Quality” studied the effect of 1-year work at the solar thermal power plants on male sperm quality. The topic is particularly important given the increasing number of solar thermal power plant. However, I have several concerns about the study.

1.      Could you provide more details about the nature of the worker’s job?? There are many possibilities, such as work involving  solar field maintenance, maintenance and repair, or operations and monitoring. Depending on their roles, workers could have direct or indirect contact with the thermal fluid used to transfer heat from solar plate to the power block or they might not have any contact with fluid at all. Additionally, part of the works are seasonal.

2.      As the authors mentioned in the Introduction “The leakage and degradation of these oils forms certain toxic species, including different types of PAH”, so this process takes time and is not typically associated with activities like installation. References 2 and 3 discuss the association between PAH and male infertility, while references 7 and 8 describe the association between male infertility and the exposure to cooking oil fumes or antifouling chemicals. These publications did not confirmed the association between infertility and the solar thermal power plants.

3.      Additionally, since the fluid used could be oil or molten salt, the main components of the fluid should be described in detail.

4.      In my opinion, the title should be changed because the authors did not specifically address the effect of  working at a solar thermal power plant. Given that exposure time was only 1 year and the study involved a small group, this study should be considered  as preliminary. Only 32 workers were at a solar thermal plant. Additionally, the p-value for abnormal morphology between exposed and non-exposed workers was nearly significant (p=0.052).

5.      I also believe that  “one year” of work does not constitute medium-term occupational exposure.

6.      The authors analyzed more than just oxidative stress, including the activity or concentration of antioxidant enzymes (SOD) or non-enzymatic parameters (GSH/GSSG). Therefore, the term of  “The oxidative stress parameters….” (line 80) should be corrected to “prooxidant-antioxidant balance parameters”.

1.      There are missing references in several places in the Introduction.

2.      The SOD abbreviation is explained twice (line 82 and 86).

3.       Descriptions of the methods or kits used are missing in the Material and Methods section.

4.      The studied groups should be described in more detail in the Materials and Methods section. 

5.      Units are missed in Table 2.

6.      Why did the authors repeat the results of SOD and GSH/GSSG on Figure 2?

7.      Did the authors measure the activity or the level of SOD? 

Author Response

1) Could you provide more details about the nature of the worker’s job?? There are many possibilities, such as work involving solar field maintenance, maintenance and repair, or operations and monitoring. Depending on their roles, workers could have direct or indirect contact with the thermal fluid used to transfer heat from solar plate to the power block or they might not have any contact with fluid at all. Additionally, part of the works are seasonal.

Done. It is showed at the beginning of material and methods section. We have expanded the information about the characteristics of the participating subjects (from both groups) but above all the peculiarities of the workers’ jobs (the duties and tasks of each position). Given that we have original information (in our mother tongue) about the workers’ tasks, we have added them to prove that what we say is true. We have added it to the end of this response file, naming it ANNEX 1.

This appendix shows many details of the workers’ tasks (it talks about inspection, verification, calibration and maintenance of valves, leak detection, verification of devices levels and pressures, etc.); also, the characteristics of the PPE (personal protection equipment).

What we want to say is that (except for the Engineers and Plant Managers), the workers rotated from one position to another, between areas of greater exposure to areas of lesser exposure. They did this not only as a potential prevention of occupational risks, but also to balance the effort and fatigue in relation to the jobs, since they also worked in shifts. We must emphasize that the information contained in Annex 1 is on the company own and is therefore confidential and should not be published (we provide it as clarification to the reviewer).

We mean that we could give many details in this regard, but we have tried to summarize, selecting the most important ones. In any case, we are willing to do what this reviewer tells us, and we will expand on it further if he tells us to do so.

2) As the authors mentioned in the Introduction “The leakage and degradation of these oils forms certain toxic species, including different types of PAH”, so this process takes time and is not typically associated with activities like installation. References 2 and 3 discuss the association between PAH and male infertility, while references 7 and 8 describe the association between male infertility and the exposure to cooking oil fumes or antifouling chemicals. These publications did not confirmed the association between infertility and the solar thermal power plants.

As discussed in the next question, we know the nature of Heat Transfer Fluid (HTF). In its degradation, PAHs are potentially produced. There is research on the associations between PAHs and male infertility. However, as this reviewer rightly says, the references we have included did not confirm the association between infertility and the solar thermal power plants. This is true (we agree with this reviewer) and this is because in this sense, our study is a first notice (there are no precedents in the literature). However, as we also say in question 1 of the ‘Detail comments’, we are going to include new references to better integrate the subject of our research.

3) Additionally, since the fluid used could be oil or molten salt, the main components of the fluid should be described in detail.

It is molten salt, as can even be seen on the website (https://www.exeraenergia.es/la-africana/)

We have more information about the company, but it is confidential and cannot be published. It is about HTF (Heat Transfer Fluid).

CONFIDENTIAL INFORMATION: The heat transfer fluid (HTF) is a eutectic mixture of biphenyl and biphenyl oxide. These substances undergo a significant increase in temperature (up to 350 ± 50 ºC) during the heat transfer operation of the solar thermal plant itself and, as a result, react to produce different cyclic aromatic hydrocarbons. According to the manufacturer’s data sheet, in this case the commercial HTF Dowtherm® from the company Dow®, the two components react as follows:

However, we have inserted brief information on the concept of heat transfer fluid (HTF) in the material and methods section.

4) In my opinion, the title should be changed because the authors did not specifically address the effect of working at a solar thermal power plant. Given that exposure time was only 1 year and the study involved a small group, this study should be considered as preliminary. Only 32 workers were at a solar thermal plant. Additionally, the p-value for abnormal morphology between exposed and non-exposed workers was nearly significant (p=0.052).

It is true. We have changed the title to a more objective one, as suggested by this reviewer. Furthermore, we have commented in the penultimate paragraph of the discussion (as a new added text) that due to the characteristics of this research it should be considered a preliminary study, as this reviewer rightly suggests. Before that (in the fourth paragraph of the discussion) we have also made a small modification mentioning the abnormal morphology of the exposed compared to the non-exposed.

5) I also believe that “one year” of work does not constitute medium-term occupational exposure.

We agree. As before, we have changed this (we wrote now one year) and it is now more objectively expressed. We have corrected it in all parts of the manuscript.

6) The authors analyzed more than just oxidative stress, including the activity or concentration of antioxidant enzymes (SOD) or non-enzymatic parameters (GSH/GSSG). Therefore, the term of “The oxidative stress parameters….” (line 80) should be corrected to “prooxidant-antioxidant balance parameters”.

Done. We have fixed it. Thanks for the advice.

Detail comments

  1. There are missing references in several places in the Introduction.

As stated in question 6 of these ‘Detail comments’ we submitted this manuscript four months ago and the editorial assistant of the journal Antioxidants told us that the introduction was too long. We had to reduce the length drastically and thus reduced the number of references. Now, following your indications, we have again extended the introduction and included bibliographical references.

  1. The SOD abbreviation is explained twice (line 82 and 86).

You are right. We have removed the second explanation. Thank you very much for pointing this out.

  1. Descriptions of the methods or kits used are missing in the Material and Methods section.

Done. We have specified this information.

  1. The studied groups should be described in more detail in the Materials and Methods section.

Done. We have tried to be more explicit. To this end, we have expanded the information concerning the characteristics of the participating subjects (from both groups) but above all the peculiarities of the jobs of the group of subjects exposed. Since we have original information (in Spanish language) about the workers’ positions, we have added them to prove that what we say is true. We have added it to the end of this response file, naming it ANNEX 1, as we said previously (in the first answer).

This appendix shows many details of the workers’ tasks (it talks about inspection, verification, calibration and maintenance of valves, leak detection, verification of devices levels and pressures, etc.); also, the characteristics of the PPE (personal protection equipment).

What we want to say is that (except for the Engineers and Plant Managers), the workers rotated from one position to another, between areas of greater exposure to areas of lesser exposure. They did this not only as a potential prevention of occupational risks, but also to balance the effort and fatigue in relation to the jobs, since they also worked in shifts.

We mean that we could give many details in this regard, but we have tried to summarize, selecting the most important ones. In any case, we are willing to do what this reviewer tells us, and we will expand on it further if he tells us to do so.

We must emphasize that the information contained in Annex 1 is on the company own and is therefore confidential and should not be published (we provide it as clarification to the reviewer).

  1. Units are missed in Table 2.

Done. We have written this information next to each parameter, as it was previously explained at the bottom of the table. We have deleted it from the table footer and placed it inside each cell.

  1. Why did the authors repeat the results of SOD and GSH/GSSG on Figure 2?

We did this because after we submitted this manuscript, the editorial assistant from MDPI Group (Miss Phoebe Liu, on May 6th, 2024) told us that we needed to expand the information about the results, among other instructions, such as that we should reduce the length of the introduction to half of what we initially sent. If this reviewer asks us to remove this information, we are willing to remove it, but we do not want to contradict anyone.

  1. Did the authors measure the activity or the level of SOD?

We measured SOD activity. Therefore, we have corrected this concept in all parts of the manuscript (including the abstract). Thank you very much for spotting it.

Reviewer 2 Report

This is an interesting study in which the effects on male reproduction parameters (in ejaculates) were compared between a group of men not working in the environment of a solar plant and a group of men that worked for one year at a solar plant.

As was indicated in the introduction working at a solar plant may involve contact with polycyclic aromatic hydrocarbons (PAH). In this study the men that were working in the solar plant for one year had a significant lower SOD activity and a lower GSH/GSSG ratio as well as a higher amount  of leukocytes when compared to the group of men not working a solar plants. Other semen parameters such as volume and  pH, amount of sperm, motility, morphology and viability of sperm were similar between the two groups.

Major concern of this study is that working or not working in a solar plant was the only discrimination between the groups.

1.The authors have described that a number of  factors may lead to exposure of PAH. It is possible that differences between the two groups tested is not due to working at a solar plant but by biased life styles between the two groups (for instance more cigarette smoking by one of the two groups or more alcohol consumption more BBQ habits, more use of lubrication oils etc.).

2. See point 1. it is also not clear whether the workers at the solar plants actually are actually exposed to PAH and to which levels (also compared with the other group). 

In my view a parameter of actual PAH levels in the "unexposed" versus "exposed" groups must ne determined (probably best is to measure PAH levels in blood but alternatively seminal plasma levels can also be determined). 

See above.

Author Response

1) The authors have described that a number of factors may lead to exposure of PAH. It is possible that differences between the two groups tested is not due to working at a solar plant but by biased life styles between the two groups (for instance more cigarette smoking by one of the two groups or more alcohol consumption more BBQ habits, more use of lubrication oils etc.).

This may be true, although in part. As we have stated in the material and methods section, we were asked about tobacco use. We did not do this because the number of smokers was very small (3 in the exposed group and 5 in the non-exposed group); as the number was insignificant, we did not take it into account. However, it is true that aspects such as diet did not constitute a group of variables that were studied in depth. Therefore, if this reviewer does not mind, we have commented on this circumstance at the end of the discussion, when we set out the limitations of our study. We appreciate this perspective from the reviewer.

2) See point 1. it is also not clear whether the workers at the solar plants actually are actually exposed to PAH and to which levels (also compared with the other group).

We think that they potentially are. In fact, the idea for this study was first suggested by the company for this very reason. Confidentially, we knew the composition of the heat transfer fluid (HTF) and therefore it was necessary to know whether the inhalation of fumes could be harmful in the medium or long term for workers. When HTF degrades, it produces PAH, as can be read in the following information about the composition of this fluid [we ask that this information not be published because it is confidential].

CONFIDENTIAL INFORMATION: The heat transfer fluid (HTF) is a eutectic mixture of biphenyl and biphenyl oxide. These substances undergo a significant increase in temperature (up to 350 ± 50 ºC) during the heat transfer operation of the solar thermal plant itself and, as a result, react to produce different cyclic aromatic hydrocarbons. According to the manufacturer’s data sheet, in this case the commercial HTF Dowtherm® from the company Dow®, the two components react as follows:

Anyway, we have inserted brief information on the concept of heat transfer fluid (HTF) in the material and methods section. Perhaps by doing so we are providing indirect information about the harmful potential of HTF degradation.

3) In my view a parameter of actual PAH levels in the "unexposed" versus "exposed" groups must be determined (probably best is to measure PAH levels in blood but alternatively seminal plasma levels can also be determined).

We agree. This would have been ideal. However, although we knew the nature of the heat transfer fluid (HTF) and the possible or potential degradation routes, what no one knew beforehand was what specific metabolites the HTF could be transformed into upon degradation. We did not have techniques for determining pollutants in ambient air. Therefore, we also did not have any optimised technique for determining these metabolites in human biological samples. The initial idea of this study is that if changes in some parameters of the workers were observed, further research should be done precisely to determine specific compounds in biological samples of the workers and thus have specific biomarkers of toxicity due to occupational exposure. This could be done at a later stage (presumably using mass spectrometry techniques, which are not currently available), but only after the results of this research are known.

We welcome this comment from the reviewer and if you do not mind, we will include a mention of this idea at the end of the discussion, when we talk about where future research should be conducted.

Round 2

Reviewer 1 Report

In the current version I accept the paper

In the current version I accept the paper